# Life Course Neighborhood Deprivation Effects on Body Mass Index: Quantifying the Importance of Selective Migration

**DOI:** 10.3390/ijerph18168339

**Published:** 2021-08-06

**Authors:** Emily T Murray, Owen Nicholas, Paul Norman, Stephen Jivraj

**Affiliations:** 1Department of Epidemiology and Public Health, University College London, London WC1B 7HB, UK; Stephen.jivraj@ucl.ac.uk; 2Department of Statistical Science, University College London, London WC1B 7HB, UK; o.nicholas@ucl.ac.uk; 3School of Geography, University of Leeds, Leeds LS2 9JT, UK; p.d.norman@leeds.ac.uk

**Keywords:** neighborhood effects, neighborhood deprivation, Townsend index, body mass index, health selection, longitudinal, birth cohort

## Abstract

Neighborhood effects research is plagued by the inability to circumvent selection effects —the process of people sorting into neighborhoods. Data from two British Birth Cohorts, 1958 (ages 16, 23, 33, 42, 55) and 1970 (ages 16, 24, 34, 42), and structural equation modelling, were used to investigate life course relationships between body mass index (BMI) and area deprivation (addresses at each age linked to the closest census 1971–2011 Townsend score [TOWN], re-calculated to reflect consistent 2011 lower super output area boundaries). Initially, models were examined for: (1) area deprivation only, (2) health selection only and (3) both. In the best-fitting model, all relationships were then tested for effect modification by residential mobility by inclusion of interaction terms. For both cohorts, both BMI and area deprivation strongly tracked across the life course. Health selection, or higher BMI associated with higher area deprivation at the next study wave, was apparent at three intervals: 1958 cohort, BMI at age 23 y and TOWN at age 33 y and BMI at age 33 y and TOWN at age 42 y; 1970 cohort, BMI at age 34 y and TOWN at age 42 y, while paths between area deprivation and BMI at the next interval were seen in both cohorts, over all intervals, except for the association between TOWN at age 23 y and BMI at age 33 y in the 1958 cohort. None of the associations varied by moving status. In conclusion, for BMI, selective migration does not appear to account for associations between area deprivation and BMI across the life course.

## 1. Introduction

Over the past 20+ years, many studies have investigated whether where people live influences their health and well-being [1]. This has included the investigation of numerous characteristics of the socioeconomic, built and social environments around individual’s residences, measured at various scales, boundaries and data sources [2]. The health outcomes linked to characteristics of places are also numerous, with the most commonly studied including physical (e.g., cardiovascular, disability and body mass index), mental (e.g., depression, anxiety) and behavioral (e.g., smoking, physical activity and alcohol use) [1,3,4]. Taken together, the literature suggests that residing in an area with some adverse environmental exposures, particularly socioeconomic, has a small but consistent association with worse health [3,5,6], usually even after adjusting for characteristics of the individuals who reside in the area.

However, very few studies examining place and health effects have incorporated a life course perspective [5]. This is problematic for a number of reasons, with the first being that health, and many risk factors for health, either accumulate over the life course and/or have long lag times between initial exposure and manifestation [7]. It is also well documented that there is a high correlation in the types of areas that individuals reside in over the life course, particularly for area deprivation [8,9,10]. Therefore, if associations are seen between area characteristics and health at one point in time, it is unknown when in the life course area could have been important for the particular health outcome.

Cumulative advantage theory also states that individuals born into advantageous or disadvantageous circumstances, such as a more deprived neighborhood, are prone to trajectories of increasing or decreasing advantage over the life course, resulting in growing health inequalities as cohorts of people age [11]. This plays out in the migration literature by people who move tending to be healthier than people who do not move [12], although there is also evidence that individuals who move to more deprived neighborhoods tend to have poorer mental and physical health [13,14,15,16,17]. Whether this is due to other processes related to downward social mobility, e.g., negative life events [18] or indirect health selection processes [19] is unclear. What is clear is that these life course migration patterns are highly selective, resulting in people with similar levels of health tending to cluster as they age [20]. It is important to the entire field of health and place to determine whether area effects on health do exist, or associations are entirely due to health selection over the life course.

In this article, we attempt to disentangle life course associations between area characteristics and health from residential health selection. Specifically, we analyze prospective longitudinal data from two British Birth Cohorts, 1958 and 1970, using the example of area deprivation and body mass index (BMI) across the life course and structural equation modelling (SEM) to assess the following: (i) whether area deprivation is associated with BMI across the life course, (ii) whether BMI selects individuals into more or less deprived areas, (iii) whether area deprivation effects on BMI are independent of residential selection by BMI and (iv) whether these area deprivation and health selection effects only occur in movers or non-movers.

## 2. Materials and Methods

This paper uses neighborhood data from the 1971–2011 Censuses linked to two British birth cohort studies, the 1958 National Child Development Study (NCDS) [21] and the 1970 British Cohort Study (BCS70) [22]. NCDS and BCS70 comprised all births, more than 17,000, in a single week in the respective baseline years (i.e., 1958 and 1970). Respondents who had immigrated to Britain were added to each sample between birth and age 16. We use the complete study sample comprising 18,558 cohort members in NCDS and 18,639 cohort members in BCS70. Data were taken from birth and follow-ups at ages 11 (1969), 16 (1974), 23 (1981), 33 (1991), 42 (2000) and 55 (2013) in NCDS and in BCS70 at ages 16 (1986), 26 (1996), 34 (2004), and 42 (2012). Data were collected through face-to-face interviews with respondents, and their parents when respondents were children. It is not currently possible to link census data before age 16 in NCDS and age 10 in BCS70 because address information is not geocoded for these study sweeps. Nonetheless, we do use information on individual circumstances of study members from birth in this study.

The outcome is life course body mass index (BMI) (kg/m^2^), which was derived in each study from measured or self-reported weight and height obtained at the following ages: 1958 NCDS: 16, 23 *, 33, 42 * and 50 y; 1970 BCS: 16 *, 26 *, 34 * and 42 * y (* self-report). The main protocol differences and methods used to harmonize height and weight have been described elsewhere [23]. Briefly, all measures were converted to metric units and implausible values were removed. BMI was also ascertained at ages 7, 11 and 44 in NCDS and at ages 10 and 30 in BCS, but these measurements were dropped from the analysis because addresses had not been geocoded prior to age 16 and to keep standard intervals of 8–10 years between measurement periods.

We linked NCDS and BCS70 individuals to Townsend deprivation index scores at the lower super output areas (LSOAs) level at each census, 1971–2011. The Townsend index is calculated by taking the average of z-score for the following census-based deprivation measures: no car access, non-home ownership, unemployment and household overcrowding [24]. All neighborhood boundaries are set at 2011 census LSOAs. LSOAs are derived using census outputs with the intention of creating evenly size population areas for which neighborhood statistics are disseminated by the Office for National Statistics (ONS)and other government bodies. They contain 1500 residents, on average. LSOAs were not designed specifically to identify neighborhoods imagined by the people who live in them, rather the main requirement is to adhere to population size constraints while respecting physical borders such as roads, railways and waterways and larger administrative boundaries, such as Local Authority Districts. Since census geographies change over time, the data for these variables are apportioned from the boundary systems used for each previous census to the 2011 Census LSOA definitions [25]. The composite index provides a relative measure of neighborhood deprivation at each census and enables comparison over time. We fixed neighborhoods over time by keeping constant their boundaries by allocating NCDS and BCS70 respondents into 2011 LSOA boundaries at ages 16, 23, 33, 42 and 55 in NCDS and ages 16, 26, 34, and 42 in BCS70. More information on the spatial linkage has been published elsewhere [26,27].

Residential mobility of cohort members was collected at each study wave, albeit asked in slightly different ways at some data collections. For each two consecutive data collections, for both cohorts (NCDS: ages 16 to 23, ages 23 to 33, ages 33 to 42 and ages 42 to 55; BCS: ages 16 to 26, ages 26 to 34 and ages 34 to 42), we re-coded residential mobility responses to obtain a consistent dichotomous measure of moved (YES = 1/NO = 0) since last study wave. For example, 1958 cohort member responses to the age 23 sweep question of “Number of places lived since 16?” was re-coded to: ‘moved (1)’ if greater than 1 address indicated and ‘not moved (0)’ if 0 or 1 address was indicated. Details of all re-coding are available in Appendix A.

The statistical analysis first consisted of descriptive statistics, by gender, calculated from imputed data using Analysis of Variance (ANOVA) for continuous variables and the chi-square statistic for categorical variables. Based on results from von Hippel’s how_many_imputations Stata 15 (Statacorp LLC, College Station, TX, USA) command [28], missing data were imputed using 50 data sets, obtained through the multiple imputation program in Mplus 7 [29]. The imputation models included the outcome and all predictors; plus auxiliary variables predictive of missingness (childhood social class, child health, birth weight and birth gender).

To examine the importance of selective migration to BMI across the life course, we used structural equation modelling (SEM) in Mplus 7 [29]. This enabled us to fit multiple mediator models with a combination of binary and continuous variables. Estimates were calculated using a percentile bootstrap applied to each imputed data set [30,31], with overall estimates calculated using Rubin’s rule [32]. Within models, all estimates are mean differences, with the outcome either mean BMI or mean Townsend score, depending on the path specified. All models were constructed to include the following theory-driven pathways: (i.) a direct ‘area tracking’ pathway (e.g., for BCS70: Area deprivation at age 16 was associated with Area deprivation age 26, Area deprivation at age 26 was associated with Area deprivation age 34, and Area deprivation at age 34 was associated with Area deprivation age 42) and (ii.) a direct ‘BMI tracking’ pathway (e.g., BMI at age 16 was associated with BMI at age 26, BMI at age 26 was associated with BMI at age 34, and BMI at age 34 was associated with BMI at age 42).

The Analysis Proceeded in Four Parts:

In the first part, we wanted to examine whether area deprivation was related to BMI across the life course. This model, henceforth known as the ‘area deprivation only’ model (see Figure 1, model A), consisted of all ‘BMI tracking’ and ‘area tracking’ pathways in the specified cohort, plus direct paths leading only from area deprivation to BMI in future (e.g., area deprivation at age 16 associated with BMI at age 26). Second, we wanted to examine whether health status, here using BMI, was related to higher or lower area deprivation at the next sweep, henceforth referred to as the ‘residential selection only’ model (see Figure 1, model B). For this model, only direct paths between BMI and area deprivation at the next sweep were added to the ‘BMI tracking’ and ‘area tracking’ paths (e.g., BMI age 16 associated with area deprivation at age 26). Third, to assess whether area deprivation effects on BMI could be explained by life course residential selection, we added all paths from the ‘residential selection only’ model to the ‘area deprivation only’ model (see Figure 1, model C).

Fourth, effect modification by residential mobility was tested to determine if area deprivation and/or health selection paths only occurred for cohort members who had moved or not moved. Separately for each association, for each study interval (i.e., 1970 cohort: ages 16 to 26, ages 26 to 34 and ages 34 to 42), a path between moved over the last interval (MOVED) and the appropriate outcome and an interaction term between the association and MOVED were added to the combined model. For example, for the question of whether the association between area deprivation at age 16 and BMI at age 26 occurred only in the 1958 cohort members who had moved or not moved from age 16 to 26, paths of MOVED between ages 16 and 26 years (MOVED16_26) and BMI at age 26 y and an interaction term of MOVED16_26 * TOWN16 (Area deprivation at age 16) were added to the full model. This resulted in a total of 14 separate interaction terms tested (7 intervals * 2 studies). If effect modification by residential mobility was not indicated, only paths between MOVEDxx_xx->BMIxx and MOVEDxx_xx->TOWNxx were included in the final model.

For all models, fit was assessed by a modified Hooper D., et al. method [33]: the chi-square statistic (χ^2^), root mean square error of approximation (RMSEA), comparative fit indices (CFI), and the standard root mean square residual (SRMR). Good overall model fit was indicated by a higher χ^2^, RMSEA ≤ 0.06, CFI ≥ 0.95, and SRMR ≤ 0.05. Sensitivity analysis was also conducted for effects of multiple testing by re-running the analysis with Bonferroni corrections.

## 3. Results

The distributions of sample characteristics are summarized in Table 1, including the unweighted N’s from the non-imputed data. In both NCDS and BCS, average BMIs increased with age, with mean values higher for the younger cohort at equivalent ages. In the NCDS sample, the average Townsend scores of cohort members were slightly worse (above 0) than the national average for ages 16 and 23, but then improve through ages 33, 42 to 55. The BCS sample respondents showed a similar pattern, with the exception they were slightly better than the national average at age 16. In both cohorts, the majority of respondents moved in their 20s and 30s with mobility rates declining with age.

Table 2 shows the direct effects and model fit for the three specified area deprivation and BMI life course models (model 1: area deprivation only, model 2: health selection only and model 3: both included) for the 1970 cohort. In all three models, both BMI and area deprivation tracked across all sweeps examined in this analysis. For example, a 1 unit higher BMI at age 16 was associated with a 0.70 (95% confidence interval 0.68, 0.72) higher BMI at age 26 years. In the area deprivation only model (1), effects of area deprivation on BMI at the next data collection were small but apparent for all three study intervals. Health selection by BMI was only weakly apparent in the last interval, showing that a 1 unit higher BMI at age 34 was associated with a 0.01 (95% CI: 0.00, 0.02) higher Townsend score at age 42. Associations of area deprivation with BMI were unchanged, and model fit statistics hardly changed, by the inclusion of the health selection pathways in the area deprivation only model (model 3). Adjusting for multiple tests attenuated all area deprivation and health selection effects in this cohort (see Appendix A).

Results were generally similar for the 1958 cohort (Table 3). Once again, both BMI and area deprivation tracked across the life course, with this cohort showing that the tracking continues into mid-life (ages 42 to 55). The area deprivation model again showed that a higher Townsend score was associated with BMI at the next sweep, with the exception of the association between area deprivation at age 23 (TOWN23) and BMI at age 33 y. In contrast to the 1970 cohort, effects of BMI on area deprivation at the next interval (health selection) were more consistent, with only the paths not significant that of the BMI at age 16 y and area deprivation at age 23 (TOWN23). However, mean differences were once again small and did not explain associations of area deprivation on BMI (model 3). After adjusting for multiple testing, only area deprivation at ages 16 and 42, and health selection between BMI at age 23 y and area deprivation at age 33 (TOWN33) confidence intervals did not contain 0 (see Appendix A).

For both cohorts, moving between a study interval did not modify any associations between area deprivation and BMI at the next interval or any associations between BMI and area deprivation at the next interval (*p*-values for interaction tests shown in Appendix A). As well, adding all residential mobility life course pathways between moving over an interval and both BMI and area deprivation (e.g., moving between age 16 and 23, BMI at age 23 and the interaction between the two variables) improved model fit substantially, so full models include pathways of life course BMI tracking, area deprivation tracking, area deprivation effects on BMI, BMI health selection and residential mobility pathways to BMI and area deprivation.

Figure 2 and Figure 3 display the full models. The addition of residential mobility paths did not alter previously examined pathways, so only the additional MOVED pathways will be discussed (leading away from the large arrows). Residential mobility effects on both BMI and area deprivation were not consistent across cohorts. For the BCS cohort (Figure 2), moving was not associated with BMI for any of the three intervals. As well, only moving between ages 16–26 y and ages 26–34 y were associated with area deprivation (TOWN) at the end of each interval, although the former was a positive association and the latter a negative association. In NCDS (Figure 3), cohort members who moved between the three earliest intervals had on average lower BMIs at the end of each interval. For example, moving between ages 16 and 23 was associated with 0.3 lower BMI (95%CI: 0.2, 0.4) at age 23 y. While only moving during the three latest NCDS intervals was associated with lower (better) Townsend scores at the end of the interval.

## 4. Discussion

Using data from two large, nationally representative UK birth cohort studies, we were able to show that across the life course, individuals who resided in more deprived neighborhoods generally had slightly higher BMIs at the next data collection. Health selection relationships, defined here as BMI being associated with area deprivation at the next interval, were tiny and less consistent across the life course, and did not explain relationships between area deprivation and BMI. As well, we also determined that neither the relationship between area deprivation on health or health selection by BMI, at any point in the life course, varied by whether the cohort member had moved or not over the past interval.

The finding that a socioeconomic area deprivation measure was weakly associated with BMI at the next study wave is consistent with previous literature [34]. Our study is able to additionally show that this relationship is generally consistent across late adolescence and adulthood, even when taking into account prior tracking of BMI and area deprivation (at least from age 16). The one exception occurred across the age 23 to age 33 waves in NCDS. Over this interval, the cohort underwent large changes in area deprivation exposure: from much worse to much better than the national average in a 10 year period. This is likely to be due to residential mobility, and not changes in areas, as 87% of the cohort moved at least once and the correlation between area deprivation in 1981 (age 23) and 1991 (age 33) was 0.91 [16,27].

Evidence for residential health selection by BMI was negligible and inconsistent across the life course and birth cohorts. In both cohorts, there was no evidence of this type of health selection across late adolescent and early adulthood years, with BCS barely showing evidence of selection between ages 34 and 42 years. It was not surprising then that area deprivation effects on BMI were not explained by residential health selection by BMI earlier in life. Our findings are similar to results from the GLOBE study (in Eindhoven, The Netherlands) [14] and the Northern Swedish Cohort (Lulea City, Sweden) [9]. We improve on these studies by showing similar findings in two nationally representative cohorts. Their studies do, however, show stronger evidence of health selection for other health measures (self-rated health and 2+ chronic diseases) [14] and some health behaviors (smoking and physical activity) [10]. It is possible then that future studies assessing relationships between area deprivation and other health outcomes might be explained by direct health selection.

The most striking aspect of this study was the high degree of tracking over the life course in area deprivation. A 1 standard deviation higher Townsend score at one age was associated with between 0.44 (ages 23 to 33 in NCDS) and 0.96 (ages 26 to 34 in BCS) average higher Townsend scores at the next age. Tracking by area deprivation is a well-documented phenomenon in the residential social mobility literature, including the limited studies that have investigated direct health selection effects [9,14,35,36]. By directly comparing the two cohorts, we were able to show here that tracking by area deprivation appears to be stronger in the younger cohort (BCS). At the same time, tracking of BMI over the life course is weaker. This analysis should be repeated in younger cohorts, to see if these lifetime geographics inequalities are being repeated in younger generations.

The main strength of this study is the availability of repeated prospective collection of nationally representative area residential addresses, and individual health and social data, across the life course. The additional linkage of harmonized area deprivation data across five decades creates a unique and powerful data set to be able to directly model and examine potential health selection bias. An additional strength was the use of structural equation models, which allowed us to simultaneously model the processes of area effects, area tracking, health selection and mobility over the life course. While each of these processes varies in their strength of contribution to inequalities in BMI, all are needed to understand why correlations between neighborhood deprivation and health outcomes change as people age.

There were a number of limitations in this study. First, we only had cohort data up until age 55 in the 1958 cohort and age 42 in the 1970 cohort. Previous research has shown that health inequalities widen as people age, peaking in mid-life [37]. This was true in our data as well, with the largest (albeit still small) association of area deprivation and BMI occurring during the latest examined wave. It is also unknown whether direct health selection by BMI might be larger at older ages, particularly at ages immediately following retirement [38]. Second, area deprivation linkage could only be conducted from age 16 for NCDS and age 10 for BCS70, as addresses from earlier sweeps of the cohorts have not been geocoded. These limitations could be examined in future if further older age sweeps and childhood area linkages became available. Third, it is possible that our examination of moderation by residential mobility could be under-powered through dichotomizing moving status, but this will need to be tested in other data sets with more consistent collection of mobility data over time. Fourth, as in any longitudinal study, attrition occurred across both samples [39,40]. However, attrition bias is most likely to create underestimates of neighborhood effects on health over time, given that cohort members residing in the most deprived neighborhoods, with the highest BMIs, would be the most likely to leave this study; both through non-response and mortality.

## 5. Conclusions

We show here, with two nationally representative birth cohorts, that direct health selection it unlikely to be driving relationships between area deprivation and health; at least for body mass index. The relationships between area deprivation and BMI themselves were consistent across the life course but small. Whether these area effects are causal is unclear. What is important is the evidence that geographic inequalities in BMI are apparent from at least adolescence and that they track strongly across the life course. Future policies and interventions are required to break these lifelong and cross-generational health inequalities.

## Figures and Tables

**Figure 1 ijerph-18-08339-f001:**
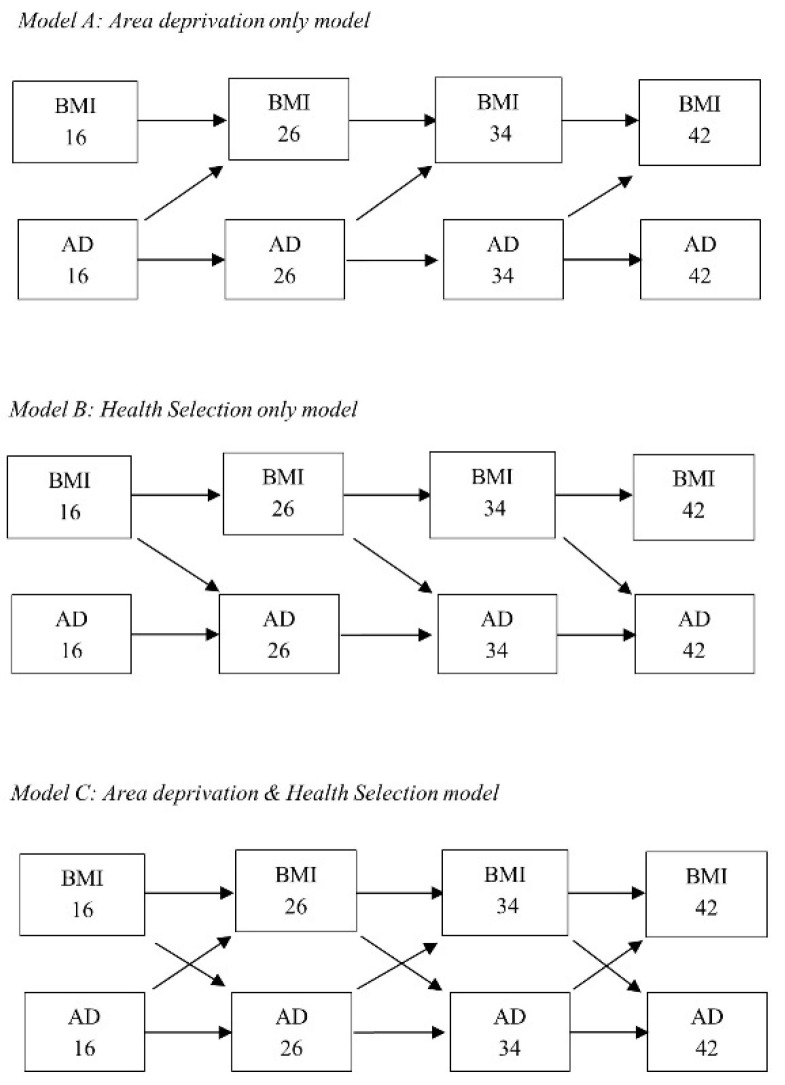
Hypothesized models linking local area deprivation (AD) and body mass index (BMI) across the life course. Model A (area deprivation only) includes only paths from area deprivation to BMI at the next interval, Model B (health selection only) includes only paths from BMI to area deprivation at the next interval and Model C includes all paths from Models A and B.

**Figure 2 ijerph-18-08339-f002:**
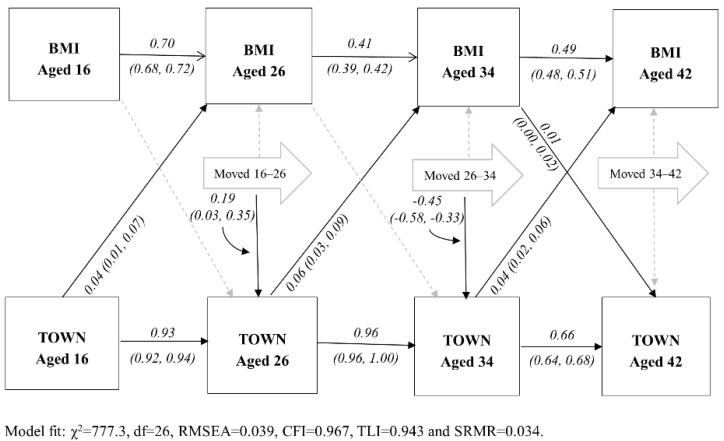
Structural equation model with both area deprivation, health selection and mobility paths across the life course: 1970 British Cohort (*n* = 18,639). Dotted line indicates non-significant paths (*p* > 0.05). BMI = body mass index; TOWN = Townsend area deprivation score.

**Figure 3 ijerph-18-08339-f003:**
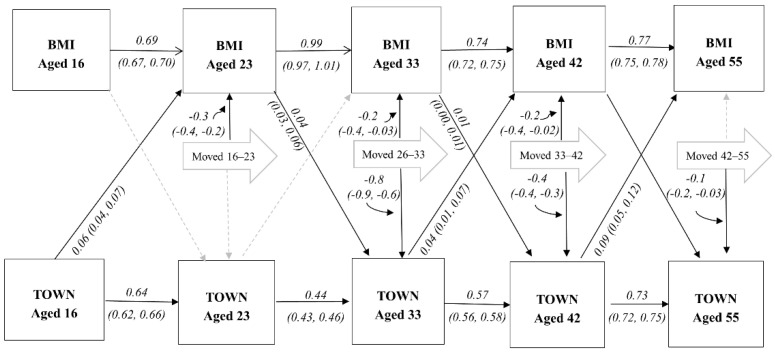
Structural equation model with both area deprivation, health selection and mobility paths across the life course: 1958 British Cohort (*n* = 18,555). Dotted line indicates non-significant paths (*p* > 0.05). BMI = body mass index; TOWN = Townsend area deprivation score.

**Table 1 ijerph-18-08339-t001:** Sample characteristics, National Child Development Study (NCDS) and 1970 British Cohort Study (BCS).

	NCDS (*n* = 18,555)			BCS (*n* = 18,639)		
	Age	Mean (SD)	Missing (%)	Age	Mean (SD)	Missing (%)
Body mass index:						
Adolescence	16	20.6 (3.68)	0.26	16	21.1 (4.51)	0.46
Early adulthood	23	22.7 (3.95)	0.23	26	23.9 (6.28)	0.56
Early mid-adulthood	33	25.1 (5.99)	0.41	34	26.0 (6.96)	0.41
Mid mid-adulthood	42	26.1 (7.08)	0.40	42	27.0 (7.51)	0.40
Late mid-adulthood	55	27.7 (8.72)	0.51	-	-	-
Area deprivation (Townsend):						
Adolescence	16	0.22 (3.68)	0.34	16	−0.20 (3.96)	0.21
Early adulthood	23	0.44 (3.81)	0.21	26	0.05 (4.10)	0.40
Early mid-adulthood	33	−0.27 (3.95)	0.35	34	−0.30 (3.28)	0.37
Mid mid-adulthood	42	−0.59 (3.13)	0.26	42	−0.58 (3.69)	0.45
Late mid-adulthood	55	−0.83 (3.27)	0.36	-	-	-
Moved between interval:		%				
Interval 1	16 to 23	0.81	0.31	16 to 26	0.83	0.56
Interval 2	23 to 33	0.87	0.44	26 to 34	0.79	0.63
Interval 3	33 to 42	0.50	0.35	34 to 42	0.69	0.51
Interval 4	42 to 55	0.29	0.43	-	-	-

**Table 2 ijerph-18-08339-t002:** Direct effects modelled using structural equation modelling and multiple imputation (1970 cohort, *n* = 18,639).

	Model 1 Area Deprivation Only		Model 2 Health Selection Only		Model 3 Both	
	Mean Difference	CI	Mean Difference	CI	Mean Difference	CI
***Fixed effects***						
Body mass index (BMI) tracking:						
BMI16 -> BMI26	0.70	0.68, 0.72	0.70	0.68, 0.71	0.70	0.68, 0.72
BMI26 -> BMI34	0.41	0.39, 0.42	0.41	0.39, 0.42	0.41	0.39, 0.42
BMI34 -> BMI42	0.49	0.48, 0.51	0.49	0.48, 0.51	0.49	0.48, 0.51
Townsend (TOWN) area deprivation tracking:						
TOWN16 -> TOWN26	0.93	0.92, 0.94	0.93	0.92, 0.94	0.93	0.92, 0.94
TOWN26 -> TOWN34	0.98	0.96, 1.00	0.98	0.96, 1.00	0.96	0.96, 1.00
TOWN34 -> TOWN42	0.66	0.64, 0.68	0.66	0.64, 0.68	0.66	0.64, 0.68
Area deprivation effect on BMI:						
TOWN16 -> BMI26	0.04	0.01, 0.07	-	-	0.04	0.01, 0.07
TOWN26 -> BMI34	0.06	0.03, 0.09	-	-	0.06	0.03, 0.09
TOWN34 -> BMI42	0.04	0.02, 0.06	-	-	0.04	0.02, 0.06
BMI effect on area deprivation:						
BMI16 -> TOWN26	-	-	0.00	−0.02, 0.02	0.00	−0.02, 0.02
BMI26 -> TOWN34	-	-	0.01	−0.01, 0.02	0.01	−0.01, 0.02
BMI34 -> TOWN42	-	-	0.01	0.00, 0.02	0.01	0.00, 0.02
***Model fit***						
χ^2^ (df)	712.4 (17)		736.9 (17)		671.35 (14)	
RMSEA	0.047		0.048		0.050	
CFI	0.968		0.966		0.970	
TLI	0.950		0.958		0.942	
SRMR	0.042		0.047		0.041	

Abbreviations: CI, confidence interval; BMI, body mass index; TOWN, Townsend; χ^2^, chi-square statistic; df, degrees of freedom; RMSEA, root mean square error of approximation; CFI, comparative fit indices; SRMR, the standard root mean square residual.

**Table 3 ijerph-18-08339-t003:** Direct effects modelled using structural equation modelling and multiple imputation (1958 cohort, *n* = 18,555).

	Model 1 Area Deprivation Only		Model 2 Health Selection Only		Model 3 Both	
	Mean Difference	CI	Mean Difference	CI	Mean Difference	CI
***Fixed effects***						
Body mass index (BMI) tracking:						
BMI16 -> BMI23	0.69	0.67, 0.70	0.69	0.67, 0.70	0.69	0.67, 0.70
BMI23 -> BMI33	0.99	0.97, 1.01	0.99	0.97, 1.01	0.99	0.97, 1.01
BMI33 -> BMI42	0.74	0.72, 0.76	0.74	0.72, 0.76	0.74	0.72, 0.76
BMI42 -> BMI55	0.77	0.75, 0.78	0.77	0.75, 0.78	0.77	0.75, 0.78
Townsend area deprivation (TOWN) tracking:						
TOWN16 -> TOWN23	0.64	0.62, 0.66	0.64	0.62, 0.66	0.64	0.62, 0.66
TOWN23 -> TOWN33	0.45	0.43, 0.46	0.45	0.43, 0.46	0.45	0.43, 0.46
TOWN33 -> TOWN42	0.57	0.56, 0.58	0.57	0.56, 0.58	0.57	0.56, 0.58
TOWN42 -> TOWN55	0.73	0.71, 0.75	0.73	0.72, 0.75	0.73	0.72, 0.75
Area deprivation effect on BMI:						
TOWN16 -> BMI23	0.06	0.04, 0.07	-	-	0.06	0.04, 0.07
TOWN23 -> BMI33	0.02	−0.01, 0.04	-	-	0.02	−0.01, 0.04
TOWN33 -> BMI42	0.04	0.01, 0.07	-	-	0.04	0.01, 0.07
TOWN42 -> BMI55	0.09	0.05, 0.12	-	-	0.09	0.05, 0.12
BMI effect on area deprivation:						
BMI16 -> TOWN23	-	-	0.01	−0.01, 0.03	0.01	−0.01, 0.02
BMI23 -> TOWN33	-	-	0.05	0.03, 0.06	0.05	0.03, 0.06
BMI33 -> TOWN42	-	-	0.01	0.00, 0.02	0.01	0.00, 0.02
BMI42 -> TOWN55	-	-	0.01	0.01, 0.02	0.01	0.01, 0.02
***Model fit***						
χ^2^ (df)	2764.4 (31)		2752.6 (31)		2614.3 (27)	
RMSEA	0.069		0.069		0.072	
CFI	0.923		0.923		0.927	
TLI	0.890		0.891		0.881	
SRMR	0.074		0.074		0.070	

Abbreviations: CI, confidence interval; BMI, body mass index; TOWN, Townsend; χ^2^, chi-square statistic; df, degrees of freedom; RMSEA, root mean square error of approximation; CFI, comparative fit indices; SRMR, the standard root mean square residual.

## Data Availability

The data presented in this study are available on request from the Centre for Longitudinal Studies: https://cls.ucl.ac.uk/ (accessed on 16 June 2021). Data are not publicly available due to potential identifiability issues.

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
