# Peer review of "Life Course Neighborhood Deprivation Effects on Body Mass Index: Quantifying the Importance of Selective Migration"

_ijerph, 2021, doi:10.3390/ijerph18168339_

Round 1

Reviewer 1 Report

The paper presents the elegant approach to the problem of  life course associations between area characteristics and health from residential health selection. Four tasks are identified, and the methods of analyses follow them .However in general the language is very "technical" and will be difficult to understand even for reader in good background of epidemiology. 

Some terms as " moved over the last interval" (MOVED), SWEEP 1-5 (table 1)' TOWN23-> BMI33 are  not necessary and could be replaced by terms well  commonly used in observational epidemiology.

Fig 1 - is not clear. The term "ND" is not described.

Final conclusions are not bringing any new messages. Tha Authors say  "We show here, with two nationally representative birth cohorts, small but consistent  associations between area deprivation and BMI across the life course. Whether these area effects are causal is unclear"

Author Response

Reviewer 1:

The paper presents the elegant approach to the problem of life course associations between area characteristics and health from residential health selection. Four tasks are identified, and the methods of analyses follow them.

RESPONSE: Thank you for your positive comments.

However, in general the language is very "technical" and will be difficult to understand even for reader in good background of epidemiology. Some terms as " moved over the last interval" (MOVED), SWEEP 1-5 (table 1)' TOWN23-> BMI33 are not necessary and could be replaced by terms well commonly used in observational epidemiology.

RESPONSE: Text throughout the manuscript has been edited to be less technical.  This includes changing row labels in Table from ‘Sweep x’ to specific life period labels (e.g. ‘Adolescence’ replaces ‘Sweep 1’). In addition, all text containing arrow symbols have been converted to text descriptions.

Fig 1 - is not clear. The term "ND" is not described.

RESPONSE: The caption for Figure 1 now includes a more detailed description of the figure. This includes changing “ND” to “AD” in the caption and figure, to designate ‘Area deprivation’.

Final conclusions are not bringing any new messages. The Authors say "We show here, with two nationally representative birth cohorts, small but consistent associations between area deprivation and BMI across the life course. Whether these area effects are causal is unclear"

RESPONSE: We have updated the conclusion section to highlight the contribution this paper makes to improved scientific understanding of this important research topic.

Page 10, rows 320-8: “We show here, with two nationally representative birth cohorts, At least for the case of BMI, we show that that direct health selection it unlikely to be driving these relationships between area deprivation and health; at least for body mass index. The relationships between area deprivation and BMI themselves were consistent across the life course but small. Whether these area effects are causal is unclear. What is important is the evidence that geographic inequalities in BMI are apparent from at least adolescence and that they track strongly across the life course. Future policies and interventions are required to break these lifelong and cross-generational health inequalities.”

Reviewer 2 Report

Authors have well prepared their manuscript. I have some minor suggestions to enhance further the quality of this research report.

Please use the word ‘multivariable’ instead of ‘multivariate’ as you have just one dependent variable in the model.

Kindly cite R software as “R Core Team (2020). R: A language and environment for statistical computing. R Foundation for Statistical Computing, Vienna, Austria. URL https://www.R-project.org/”.

Table 5, total denominator numbers for men and women were reported in decimals (i.e., 1060.4 for men and 1056.8 for women) which looks weird. Is it weighted number?

Table 7, did authors check for any meaningful interaction effect between predictors? A sentence in the analysis plan should be added.

Author Response

Reviewer 2:

Authors have well prepared their manuscript. I have some minor suggestions to enhance further the quality of this research report.

RESPONSE: Thank you for your positive comments and suggestions.

Please use the word ‘multivariable’ instead of ‘multivariate’ as you have just one dependent variable in the model.

Kindly cite R software as “R Core Team (2020). R: A language and environment for statistical computing. R Foundation for Statistical Computing, Vienna, Austria. URL https://www.R-project.org/”.

Table 5, total denominator numbers for men and women were reported in decimals (i.e., 1060.4 for men and 1056.8 for women) which looks weird. Is it weighted number?

Table 7, did authors check for any meaningful interaction effect between predictors? A sentence in the analysis plan should be added.

RESPONSE: Upon closer inspection, we do not believe these comments apply to our paper. We have not used the word ‘multivariate’ in our manuscript, have used MPlus software for the analysis and there are only three tables in the manuscript. Has the reviewer submitted the wrong reviewer comments?

Round 2

Reviewer 1 Report

Autors responded well to mycomments. No further remarks. 

Author Response

Thank you.

Reviewer 2 Report

I really do not know how these irrelevant comments were posted in the earlier version of review, and I really apologies for that. However, my initial minor comments are listed below.

Authors that missing data was imputed using 50 data sets, obtained through the multiple imputation program in Mplus 7.29. What was the rationale for choosing 50 imputations (i.e., based on the proportion of missing or the fraction of missing)? It should be explained and supported by a citation. A simulation study conducted by Madley-Dowd et al (2019) indicated that the fraction of missing information is better as a guide to the efficiency gains from MI than the proportion of missing data (https://doi.org/10.1016/j.jclinepi.2019.02.016).

Wondering if gender information can be added in the table 1 if available. Please ignore this comment if it not possible.

In tables 2 and 3, full abbreviations should be provided in the table footnote for acronyms used in these tables.

Authors mentioned that model fit was assessed by the chi-square statistic (χ2), root mean square error of approximation (RMSEA), comparative fit indices (CFI), and the standard root mean square residual (SRMR). However, authors are suggested to mention the reason for choosing these measures only as many more measures exist in the SEM literature (https://doi.org/10.1186/s13717-016-0063-3).

Author Response

Please see the attachment, which includes the detailed the changes according to reviewer# 2 'comments.
